# Functional and Mass Spectrometric Evaluation of an Anti-Tick Antigen Based on the P0 Peptide Conjugated to Bm86 Protein

**DOI:** 10.3390/pathogens9060513

**Published:** 2020-06-25

**Authors:** Alina Rodríguez Mallón, Luis Javier González, Pedro Enrique Encinosa Guzmán, Gervasio Henrique Bechara, Gustavo Seron Sanches, Satomy Pousa, Gleysin Cabrera, Ania Cabrales, Hilda Garay, Raúl Mejías, José Raúl López Álvarez, Yamil Bello Soto, Fabiola Almeida, Osmany Guirola, Rafmary Rodríguez Fernández, Alier Fuentes Castillo, Luis Méndez, Samanta Jiménez, Alexei Licea-Navarro, Madelón Portela, Rosario Durán, Mario Pablo Estrada

**Affiliations:** 1Animal Biotechnology Department, Center for Genetic Engineering and Biotechnology (CIGB), Havana 10600, Cuba; pedroencinosa88@gmail.com (P.E.E.G.); yamil.bello@cigb.edu.cu (Y.B.S.); mario.pablo@cigb.edu.cu (M.P.E.); 2Mass Spectrometry Laboratory and GlycoLab, Department of Proteomics, CIGB, Havana 10600, Cuba; luis.javier@cigb.edu.cu (L.J.G.); satomy.pousa@cigb.edu.cu (S.P.); gleysin.cabrera@cigb.edu.cu (G.C.); falmeida@gmail.com (F.A.); 3Programa de Pós-graduação em Ciência Animal, Pontifícia Universidade Católica do Paraná (PUCPR), Paraná 80215-901, Brazil; gervasio.bechara@pucpr.br (G.H.B.); gustavoseron@hotmail.com (G.S.S.); 4Departamento de Patologia Veterinária, Faculdade de Ciências Agrárias e Veterinárias, Universidade Estadual Paulista (FCAV-UNESP), São Paulo 14884-900, Brazil; 5Synthetic Peptides Group, CIGB, Havana 10600, Cuba; ania.cabrales@cigb.edu.cu (A.C.); hilda.garay@cigb.edu.cu (H.G.); 6Instituto de Ciencia Animal (ICA), San José de las Lajas 32700, Cuba; rmejias@ica.co.cu (R.M.); jrlopez@ica.co.cu (J.R.L.Á.); 7Bioinformatic Department, CIGB, Havana 10600, Cuba; osmany.guirola@cigb.edu.cu; 8National Laboratory for Parasitology, San Antonio de los Banos 32500, Cuba; rafmary.rodriguez@lnp.art.minag.cu (R.R.F.); alier.fuentes@lnp.art.minag.cu (A.F.C.); luis.mendez@lnp.art.minag.cu (L.M.); 9Departamento de Innovación Biomédica, CICESE, Ensenada 22860, Mexico; sjimenez@gmail.com (S.J.); alnavarro@gmail.com (A.L.-N.); 10Unidad de Bioquímica y Proteómica Analítica, Institut Pasteur de Montevideo, Montevideo 11400, Uruguay; mportela@gmail.com (M.P.); rduran@gmail.com (R.D.); 11Instituto de Investigaciones Biológicas Clemente Estable (IIBCE), Montevideo 11600, Uruguay

**Keywords:** anti-tick vaccine, P0, Bm86, peptide, chemical conjugation, cross-linked peptides

## Abstract

A synthetic 20 amino acid peptide of the ribosomal protein P0 from ticks, when conjugated to keyhole limpet hemocyanin from *Megathura crenulata* and used as an immunogen against *Rhipicephalus microplus* and *Rhipicephalus sanguineus* s.l. species, has shown efficacies of around 90%. There is also experimental evidence of a high efficacy of this conjugate against *Amblyomma mixtum* and *Ixodes ricinus* species, which suggest that this antigen could be a good broad-spectrum anti-tick vaccine candidate. In this study, the P0 peptide (pP0) was chemically conjugated to Bm86 as a carrier protein. SDS-PAGE analysis of this conjugate demonstrated that it is highly heterogeneous in size, carrying from 1 to 18 molecules of pP0 per molecule of Bm86. Forty-nine out of the 54 lysine residues and the N-terminal end of Bm86 were found partially linked to pP0 by using LC-MS/MS analysis and the combination of four different softwares. Several post-translational modifications of Bm86 protein were also identified by mass spectrometry. High immunogenicity and efficacy were achieved when dogs and cattle were vaccinated with the pP0–Bm86 conjugate and challenged with *R. sanguineus* s.l. and *R. microplus*, respectively. These results encourage the development of this antigen with promising possibilities as an anti-tick vaccine.

## 1. Introduction

Ticks cause not only direct damage to infested hosts by debilitating effects but they may also act as vectors of agents that cause diseases that in many cases can be lethal [1]. The most common method to control ticks involves chemical products that cause environmental pollution and the appearance of resistant tick lineages. Tick immunological control, as an alternative to acaricides, is based on host immunizations with an antigenic protein derived from ticks. The immune response against these antigens should be able to affect the parasite life cycle, reducing the viable larva “yield” after each generation of parasites fed on vaccinated animals [2]. Many attempts to identify such antigens have been carried out by anti-tick vaccine developers after successful *Rhipicephalus microplus* immunological control was achieved by cattle vaccination with the concealed antigen Bm86. This is a glycoprotein identified by an Australian group that is located at the epithelial cell membrane of the tick gut [3].

Gavac is the trade name of the Cuban commercial vaccine against ticks based on Bm86 protein expressed by *Komagataella phaffii* (*Pichia pastoris*) yeast [4] produced at the at the Center for Genetic Engineering and Biotechnology, in Havana, Cuba [5,6]. It has been applied to more than 3 million cattle in Cuba, Venezuela, Mexico, Brazil and Colombia with an efficacy between 51 and 99% depending on the *R. microplus* tick strain. This is the largest published evidence in the world about the efficacious application of cattle tick immunologic control under field conditions [7,8,9]. Nevertheless, the identification of new antigens with high efficacy against other tick species remains a necessity to achieve effective tick control by vaccination. More recently, an antigen based on the peptide of the P0 acidic ribosomal protein of ticks from amino acid 282 to 301 (pP0) chemically conjugated to keyhole limpet hemocyanin (KLH) of *Megathura crenulata* was evaluated against *Rhipicephalus* spp. Ticks, showing around 90% of efficacy [10,11], and also against *Amblyomma mixtum* [12] and *Ixodes ricinus* ticks (manuscript in preparation). However, the pathway from concept proof in controlled experiments to the development of a commercial vaccine is a long, difficult and costly process. Selecting the right strategy to efficiently generate large quantities of antigen with suitable vaccine properties is a considerable challenge to all researchers working on this topic. In this sense, KLH is not an economically feasible carrier for pP0 development because it is obtained from its natural source and consequently is very expensive.

The aim of this study was the efficacy evaluation of the chemical conjugate of this pP0 with the Bm86 protein used as the active ingredient of Gavac^TM^. The pP0–Bm86 conjugate was obtained by using the *N*-β-(maleimidopropyloxy) succinimide ester (BMPS) as a heterobifunctional cross-linker between the primary amino groups of the carrier protein (lysine residues and the amino terminal group) with the thiol group of a Cys residue intentionally added at the N-terminal end of pP0. The average number of pP0 molecules linked to the carrier protein was determined by using SDS- PAGE analysis of the pP0–Bm86 conjugate. LC-MS/MS analysis of peptides from the conjugate digested with several specific proteases was performed for the identification of conjugation sites and post-translational modifications (PTMs) of the Bm86 protein. Finally, the pP0–Bm86 conjugate was used to immunize dogs and cattle that were subsequently challenged with *R. sanguineus* and *R. microplus* ticks, respectively, in order to determine anti-tick efficacy.

## 2. Results

### 2.1. Estimation of pP0 Molecules Linked to Bm86 Carrier Protein

SDS-PAGE analysis of Bm86 protein showed a 30 kDa broad band ranging between 83 and 113 kDa (Figure 1**,** lane II). This is the typical migration of proteins with *N*-glycosylation sites partially occupied by *N*-glycans that are heterogeneous in size [13]. After deglycosylation with PNGase-F, Bm86 migrated on SDS-PAGE as a single band with around 91 kDa of molecular mass (Figure 1**,** lane III). According to its amino acid composition, the Bm86 protein produced by our *Pichia pastoris* yeast clone should have a molecular weight around 67.8 kDa. Bovine Serum Albumin, used as a control in this experiment, migrated at 67 kDa, as expected (Figure 1, lane VI). The abnormal migration of this Bm86 protein could probably be related to a gel shifting phenomenon reported for some cytosolic and membrane proteins with a SDS affinity lower than expected [14,15]. As a consequence, these proteins with a considerably lower number of negative charges than those corresponding to their size show a reduced mobility on SDS-PAGE analysis and migrate as proteins with a higher molecular mass than expected. The trend of this Bm86 protein produced in yeasts towards self-aggregation [16] could also be considered as a possible cause of abnormal migration because the conventional sample treatment prior to SDS-PAGE analysis could be not enough to dissociate all protein–protein interactions and the high number of disulfide bonds. In fact, the Bm86 protein has 65 cysteine residues in its amino acid sequence and the uncompleted reduction of disulfide bonds could make some SDS binding domains unavailable to the detergent and cause abnormal migration in SDS-PAGE analysis [17].

The pP0–Bm86 conjugate also migrated as a diffuse band of 45 kDa width ranging between 97 and 142 kDa (Figure 1, lane IV). Meanwhile, deglycosylated pP0–Bm86 conjugate migrated as a diffuse band of 34 kDa width ranging from 93 to 127 kDa (Figure 1, lane V). According to the mass difference ranging from 2 to 36 kDa between Bm86 and pP0–Bm86 deglycosylated proteins (Figure 1, lanes III and V), the payload was estimated to range from 1 to 18 molecules of pP0 for each Bm86 molecule, considering 2 kDa as the mass of a pP0 molecule. This result suggested that the pP0–Bm86 conjugate was heterogeneous in size, probably because in the chemical conjugation the excess of added BMPS targeted all accessible lysine residues on the protein surface with the subsequent extensive cross-linking of pP0 that is also added in a molar excess in the final step of the reaction [18]. The molecular mass difference between glycosylated proteins was not considered in the pP0 payload estimation because non-protein components, such as *N*-glycans, contribute to increased protein molecular mass but do not bind SDS uniformly and may alter the molecular mass determination by SDS-PAGE analysis [15].

### 2.2. Identification of Conjugation Sites in pP0–Bm86 Conjugate

PEAKS software identified 140 MS/MS cross-linked peptides that allowed the assignment of 42 conjugation sites out of the 55 sites that could have been potentially susceptible to conjugation in the sequence of the Bm86 protein. Forty-one of them corresponded to internal lysine residues and the additional conjugation site was located at the N-terminal end of the Bm86 protein (Figure 2A, Appendix A).

Protein Prospector, StavroX and pLink software identified from 158 to 228 MS/MS spectra of type II cross-linked peptides, which were assigned to 41, 41 and 46 conjugation sites, respectively, in the Bm86 sequence (Figure 2A and Appendix A). The results of these software programs, specifically developed for the identification of type II cross-linked peptides, were highly complementary to the PEAKS results and very useful to validate them. Only three conjugation sites, located at K279, K338 and K424, were identified exclusively by PEAKS software (Figure 2B,C and Appendix A). Eight new conjugation sites, corresponding to other internal lysine residues of the Bm86 sequence, were found in addition to those found by PEAKS software with the use of further software (Figure 2B and Appendix A, K53, K241, K358, K438, K534, K537, K582 and K603). Four of them (K241, K358, K534 and K582) were coincidently identified by all three software programs. 

Conjugation sites coincidently identified by the four software programs were 68% and 84% if the combination of three software programs is considered (Figure 2B), which demonstrated the consistency of these results. An example of complementarity among the results obtained from all software programs was the assignment of K527 as a conjugation site by PEAKS software, which was supported by only one MS/MS spectrum (Appendix A, row 226, [peptide of Bm86]-(fragment of pP0) [520–528]-(1–16), 4+;). However, this identification was also supported by the other three software programs. As a consequence, four additional peptides containing the same conjugation site were also identified ([520–528]-(1–17), 4+; [524–528]-(1–16), 4+; [524–530]-(1–13), 3+; [524–530]-(1–14), 3+) and supported by 15 additional MS/MS spectra (Figure 2C, Appendix A, rows 225–235). Despite that, K305, K554, K564, and K582 were assigned as conjugation sites by the identification of only one peptide each (Figure 2C), their corresponding MS/MS spectra were coincidently assigned to intermolecular type II cross-linked peptides by three out of the four software programs used (Appendix A, rows 121, 245, 246 and 255). In contrast, only six (K279, K338, K424, K438, K537 and K603) out of the forty-nine conjugated lysine residues in the Bm86 protein sequence were supported by only one peptide and the assignment of only one software (Figure 2B,C and Appendix A). Another redundant result that supported the reliability of the 274 MS/MS spectra assigned to type II cross-linked peptides in this work was the 26.3% identification coincidence obtained with all software programs. Different combinations of three and two software programs coincidently identified 52.6% and 76.9%, respectively (Appendix A). 

Despite all this redundancy, the presence of diagnostic ions was verified manually as another validation of conjugation sites. An example of this verification is shown in Figure 3. Two electrospray mass spectrometry (ESI-MS) spectra, whose signals were detected at *m/z* 826.61, 4+ and *m/z* 936.09, 3+, were assigned by PEAKS software to two intermolecular type II cross-linked peptides (Figure 3A–D). In both peptides, the N-terminal Cys residue of the proteolytic fragment of pP0 (1–16) was linked in one case to the N-terminal end of the Bm86 protein [1–14] (Figure 3B) and in the other case to the lysine 468 in the peptide [465–475] of the Bm86 protein (Figure 3E). Intense ions at *m/z* 589.28, shown by corresponding MS/MS spectra, were assigned to y5β and y5α, respectively (Figure 3C–F). These ions were a result of peptide bond fragmentation at the N-terminal of the proline residue located at the 5th position of pP0 (1–16) (Figure 3B–E). Additionally, the ion at *m/z* 784.84, 2+ generated by the dissociation of the amide bond between the linker and the side chain of K468 of the Bm86 protein (Figure 3E) was assigned as (pP0[1–16] +linker)^2+^, confirming the previous assignment (Figure 3F).

On the other hand, the use of different proteases was decisive to obtain a high sequence coverage and mapping of conjugation sites in the Bm86 protein. Peptides that were missed by tryptic digestion were identified by tandem digestion with trypsin and V8 protease and vice versa. For example, the conjugation site located at K31 of the Bm86 protein sequence was detected in four different cross-linked peptides of the Bm86 protein [15–34], [20–34], [26–34] and [28–34] (Appendix A, rows 6–12). In the same way, due to the presence of several cleavage sites in pP0, the same proteolytic peptide of Bm86 was also found linked to different pP0 fragments, either (1–16, 1–17, 1–21) or (1–13, 1–14, 1–18 and 1–19) when trypsin or *Staphylococcus aureus* V8 protease were used, respectively (Appendix A).

Furthermore, ESI-MS, as an ionization method, also provided a valuable redundancy, because MS/MS spectra of the same cross-linked peptide carrying different charge states were coincidently detected by different software in similar retention times but at different *m/z* (Appendix A) [20]. For example, the conjugation site at K146 of the Bm86 protein sequence was identified in 10 different peptides and with different charge states in their ESI-MS spectra. Considering the contribution of the four software programs, this lysine residue was identified 42 times as a conjugated residue (Figure 2C and Appendix A, lines 57–70).

Linear peptides generated by trypsin are ionized in a gas phase mainly as doubly and triply charged ions because they have a free amino terminal end and a basic residue at the C-terminal end (either lysine or arginine). On the contrary, type II cross-linked peptides with two N-terminal ends and two carboxyl ends duplicate the potential positions to be protonated. For this reason, type II cross-linked peptides are ionized preferentially with Z ≥ 3+ and very few (less than 5%) as doubly charged ions [21]. Our study agreed with this regularity because linear peptides were almost exclusively ionized as 2+ and 3+ in 97.6% of cases, while type II cross-linked peptides had a Z ≥ 3+ in more than 97.1% of cases (Appendix A). 

In summary, 49 lysine residues out of 54 in the Bm86 protein sequence and its N-terminal end were found conjugated to pP0 with the contribution of the four software programs used to analyze MS/MS spectra (Figure 2C). However, it was also noticed that 46 out of 49 conjugated lysine residues were also found unmodified and were detected as linear peptides, suggesting their partial modification after the conjugation reaction (Appendix A, K residues written in blue). A different extension of the conjugation on each individual lysine residue is in agreement with very intense signals of detection for some types of II cross-linked peptides, while others were detected with a very poor signal-to-noise ratio in the ESI-MS spectrum (Figure 3A,D and Appendix A, rhombuses). This partial conjugation of Bm86 lysine residues could contribute to the size heterogeneity of the pP0-Bm86 conjugate, which is in good agreement with the broad band observed in the SDS-PAGE analysis of deglycosylated conjugate (Figure 1, lane V). However, the approach assumed in this work was not quantitative and did not allow the determination of the ratio between free and conjugated lysine residues for each individual site. Only three lysine residues in the Bm86 protein sequence were detected as fully conjugated (Appendix A, K102, K302, and K600 written in red). No conjugation evidence was found at K residues located at positions 180, 431, 484, 495 and 505. However, the possibility that some lysine residues of the carrier protein even conjugated to pP0 could not be identified because they were not efficiently fragmented by collision-induced dissociation (CID) should not be excluded. The collision energy profiles available in our mass spectrometer can be customized only for 2+ and 3+ ions and not for ions with charge states higher than 3+ ions, and around 68.6% of type II cross-linked peptides identified in our MS/MS spectra had Z ≥ 3+. On the other hand, some type II cross-linked peptides had molecular masses >3500 Da that exceeded the reported limit for an efficient fragmentation using CID. Better fragmentation efficiency and sequence coverage of cross-linked peptides could be obtained with the combination of several fragmentation methods (ETD, HCD, ETciD, EThcD) that have recently been proposed [21]. In the mass spectrometer where this study was performed, only collision-induced dissociation is available as a fragmentation method.

Although the identification of intermolecular type II cross-linked peptides was our main objective, linear peptides mainly detected as 2+ ions were not excluded from the LC-MS/MS analysis, which allowed the sequence verification of the Bm86 protein. Overlapping the information provided by the LC-MS/MS analysis of all proteolytic digestions, including the identification of linear and cross-linked peptides by PEAKS software, 91% of the pP0–Bm86 conjugate amino acid sequence was verified (Appendix A). None of the 5255 proteins included in the sequence database of *P. pastoris* host cell was identified in the pP0–Bm86 conjugate, confirming the purity of the Bm86 protein used in the conjugation process.

### 2.3. Post-Translational Modifications of the Bm86 Protein

All PTMs found in the ESI-MS analysis are summarized in Appendix A. Many of them are frequently reported for heterologous proteins expressed in yeast [22] and should not be associated to the chemical reaction used to obtain the pP0–Bm86 conjugate. *N*-glycosylation seems to be the most abundant PTM of the Bm86 protein, considering the number of proteolytic glycopeptides detected. The *N*-glycosylation profiling at the four potential sites on Asn 122, 163, 329 and 363 residues of the Bm86 amino acid sequence showed heterogeneous high-mannose *N*-glycans from Man_8_ to Man_12_, where Man_8_ and Man_9_ were the most abundant [23] (Appendix A). Matrix-assisted laser desorption/ionization mass spectrometry (MALDI-MS) analysis of *N*-glycan pool released by PNGase F treatment and modified with 2AB confirmed this result (Appendix A). The ESI-MS and ESI-MS/MS analysis also confirmed the *N*-glycopeptide assignments, because after treatment with PNGase-F, deamidation at the formerly *N*-glycosylated Asn residue was found. Several peptides with a 162 Da increment in their molecular mass were also found in the ESI-MS analysis. These signals were tentatively assigned to O-glycopeptides because this PTM is frequently introduced in heterologous proteins expressed in yeast [23,24]. However, the O-glycosylation site assignment based on MS/MS experimental evidence was not obvious, probably due to the instability of this PTM in the CID conditions [20]. Anyway, O-glycosylation and phosphorylation in Ser^3^ and Ser^6^, respectively, were found. These PTMs seem to be of low abundance, judging by the intensity shown by the precursor ions in the ESI-MS analysis (Appendix A)

Two peptides with formylated Lys residues were unequivocally detected (Appendix A, peptides 34 and 36). There are previous reports showing that endogenous formaldehyde is an important source of N^6^-formyl lysine and also that this modification is extended to proteins present in all cell compartments [25,26]. Formylation at lysine residues is increased in proteins in a dose-dependent manner upon exposure to formaldehyde [27]. The transformation of methanol into formaldehyde and hydrogen peroxide in the peroxisomes during methanol fermentation due to the overexpression of the alcohol oxidase (AOX) gene in *P. pastoris* may have a direct relation to the presence of this PTM in our Bm86 protein.

Deamination in several Asn and Gln residues and the oxidation to sulfoxide of Met residues were also found. These PTMs could be associated with sample preparation prior to mass spectrometry, which includes protein denaturalization after reduction and S-alkylation, basic pH treatments and a prolonged proteolytic digestion at 37 °C. These conditions could be favorable for the generation of these modifications.

Peptide [1–14] was found partially modified at the N-terminal end with acetyl, pyroglutamyl and carbamyl blocking groups (Appendix A, peptides 1–3). In particular, the carbamylation of the N-terminal end could be associated with an artifact introduced in the the purification process during the Bm86 protein extraction by using urea, because the carbamylation pathway is exclusive to higher organisms [28,29] and it has not been described for yeasts. 

This mass spectrometric characterization of the pP0–Bm86 conjugate allowed a deeper characterization of the Bm86 protein produced by a recombinant *P. pastoris* yeast, which apart from being used as a carrier protein for pP0 in this study, is the pharmaceutical active ingredient of Gavac^TM^ [8,30].

### 2.4. Efficacy of Immunization with pP0–Bm86 Conjugate against R. sanguineus and R. microplus Ticks

There were no changes in normal behavior in any of dogs and bovines during the immunization experiments. In dogs, some local inflammatory responses in the vaccine application site were evident but disappeared after two or three days without any treatment (data not shown). Immunized bovines did not show any clinical signs of disease. They increased their body weight by around 100 kg during the experiment and kept their hematological parameters within the normal limits described for the species [31] (Figure 4).

Specific antibody responses against pP0 were obtained in all vaccinated animals (dogs and bovines) with the chemical conjugates of pP0 regardless carrier protein (KLH or Bm86). No statistical differences were found at any time in the anti-pP0 titers generated by the immunized animals with the pP0–KLH conjugate compared to those immunized with pP0–Bm86 conjugate (Figure 5). The kinetics of the immune response was similar for dogs and bovines. The magnitude of these responses is not comparable between them because there were differences in the indirect ELISAs used for measuring specific antibody titers in both species. In bovines, 170 days after the first immunization, the titers against pP0 decreased to around 1000. There is evidence that anti-pP0 antibody titers below 1000 may be insufficient to affect ticks, but further studies should be designed in order to accurately establish this lower limit and the best time for booster immunizations. Specific antibody responses were also obtained against carrier proteins. In the case of anti-Bm86 titers in bovines, after 170 days, they were even above 640, which is the minimum protective titer described for the Gavac^TM^ vaccine [7]. This result agreed with the current immunization schedule for Gavac^TM^ vaccine application, which includes two initial immunizations followed by a booster, each six months apart [32].

Dog vaccination with pP0 conjugates did not show an effect on the *R. sanguineus* larval stage in this experiment. The most significant reductions, compared to the control group, were obtained on nymph and adult yields and also in mortality during the nymph molting process (Table 1). As was previously described [11], a high quantity of engorged females from pP0-vaccinated animals were unable to lay eggs and died a few days after being collected. This mortality was included in the female yield parameter that was found to be statistically different in the vaccinated groups compared to the control group. There were also statistically significant differences in the percentage of hatched eggs in the group immunized with pP0 conjugates compared to the control group. The overall efficacy calculated for pP0–KLH and pP0–Bm86 immunogens was 95% and 86%, respectively.

Morpho-histological changes observed in the ovaries of *R. sanguineus* females fed on immunized dogs could explain the differences in hatchability (Figure 6). Pre-vitellogenic oocytes in both groups that received pP0 conjugates showed irregular shapes and also cytoplasmic vacuolization, which is indicative of cell death. These effects were more frequently observed in the pP0–KLH group than in pP0–Bm86, however, this last group presented many oocytes in the pre-vitellogenic stage and mature oocytes with a significant reduction in yolk granule deposition in relation to the control group (*p* < 0.001). Scanning electron microscopy (SEM) verified that the ovaries from females fed on control dogs were more developed, presenting a significantly greater amount of mature oocytes than the ovaries of females fed on immunized groups (*p* < 0.05) (Figure 7). The ovaries of females fed on the pP0–KLH and pP0–Bm86 groups had some mature oocytes but also a large number of immature oocytes adhered to the ovary wall.

The oocyte ultrastructure from females fed on control dogs is in accordance with the first ultrastructure description of *R. sanguineus* ovaries [33] (Figure 8AA). Oocytes III presented small rounded granules of proteins and lipids. Lipid granules are more electron dense. Mitochondria with different formats and with ridges organized in parallel were observed in the cytoplasm, mainly in the perinuclear region. The nucleolus was also evident, suggesting intense synthesis activity (Figure 8(AA-B)). Numerous microvilli and a slide thick basal membrane are present in the plasmatic membrane (Figure 8(AA-A)). On the other hand, oocytes IV showed a large amount of protein granules with different sizes and electron densities distributed throughout the cytoplasm (Figure 8(AA-E)). Lipid granules were also observed in their cytoplasm. They are smaller than protein granules and are electron-lucid granules (Figure 8(AA-G)). Few mitochondria were observed in the cytoplasm, among granules, however, it was possible to observe the presence of a well-developed Golgi complex associated with an intense synthesis process (Figure 8(AA-H)). The initial chorion deposition by cell exocytose was also observed as polymerized vesicles in the extracellular space between the basal lamina and the plasmatic membrane of the cells (Figure 8(AA-F,G)). Oocytes V showed large electron-dense yolk granules, dispersed throughout the cytoplasm (Figure 8(AA-I,L)). In the peripheral region, two layers of chorion are visible. The outermost layer, or exochorion, appears with a moderated electron density and the endochorion, or the innermost layer, is thicker and it is in direct contact with the oocyte (Figure 8(AA-I,L)). 

In contrast, oocytes from females fed on pP0–KLH conjugate-immunized dogs showed a great extension of cytoplasmic degradation (Figure 8BB). Oocytes III showed deformations in the plasmatic membrane (Figure 8(BB-A)), where the absence of microvilli was also observed (Figure 8(BB-B)). These structures are responsible for the absorption of important exogenous elements used by oocytes for yolk synthesis. Intense cytoplasmic degradation at the pedicel–oocyte interface was observed in some oocytes III, suggesting that damage is coming via the pedicel (Figure 8(BB-C)). In addition, it was also possible to observe the presence of non-homogeneous yolk granules in the oocyte cytoplasm, mitochondria in the degradation process and the presence of autophagy vacuoles and myelinated figures, which are indicative of cell death (Figure 8(BB-B,D,E)). The oocytes IV presented fibrillary chorions in the peripheral region with few microvilli and degraded cytoplasm with autophagy vacuoles (Figure 8(BB-F,G)). Non-homogeneous yolk granules were also observed to be sparse in the cytoplasm and irregularly shaped (Figure 8(BB-H,I)). Oocytes V showed changes similar to those observed in oocytes IV (Figure 8(BB-J–L)).

Oocytes III, IV and V from females fed on dogs immunized with the pP0–Bm86 conjugate also show evident changes, mainly at the cytoplasm membrane and in the yolk formation, producing, as a consequence, cell degradation (Figure 8CC). Oocytes III also show an irregular plasmatic membrane with discontinuities in the basal lamina (Figure 8(CC-A)). In these regions, a large amount of autophagy vacuoles and degraded cytoplasm regions were observed (Figure 8(CC-B)). Mitochondria with a rounded shape in the degeneration process and fragmented cytoskeleton were also observed (Figure 8(CC-C,D)). In the same way, oocytes IV in this group had fibrillary chorions with few microvilli (Figure 8(CC-E,G)). Degradations and rare mitochondria were also observed (Figure 8(CC-F)). Irregularly shaped yolk granules with sizes smaller than usual were also evident in the oocyte cytoplasm. (Figure 8(CC-H)). Oocytes V showed the most significant morphological changes in this group. An accentuated fibrillary chorion was observed with a large amount of autophagy vacuoles between it and the basal lamina. Cytoplasm regions with extensive degradation were observed mainly close to large yolk granules similar to the verified changes in the ovaries of ticks treated with chemical acaricides [34]. Rounded mitochondria and autophagy vacuoles were also observed (Figure 8(CC-I–L)).

In the bovine experiment, a significant reduction in the female yield of *R. microplus* was obtained from bovines immunized with pP0 conjugates compared to the control group (Table 2). A high number of engorged females collected from bovines immunized with pP0 conjugates were unable to lay eggs and died a few days after detaching (Figure 9). This effect on female mortality was more evident in the pP0–KLH-immunized group and was included in the female yield parameter. Females from bovines immunized with pP0–Bm86 conjugate laid significantly fewer eggs than those fed on bovines in the control group. Statistically significant differences were also found in hatchability when vaccinated groups were compared to the control group. Ovary ultrastructural analysis was not performed on *R. microplus* ticks fed on vaccinated cattle, but the same changes observed in the ovaries of *R. sanguineus* ticks fed on vaccinated dogs probably occurred, taking into account the similar observed effect. The overall calculated efficacy for pP0–KLH and pP0–Bm86 conjugates on *R. microplus* ticks was 89% and 84%, respectively. The efficacy was found to be strongly correlated with specific antibody titers against vaccine antigens (Figure 10B). In the dog experiment, the efficacy was significantly correlated with specific antibody titers against pP0. There was no significant correlation between efficacy and antibody titers against Bm86 (Figure 10A).

## 3. Discussion

Many authors agree that the combination of several immunogens, involved in different physiological processes in the same preparation, could contribute to an increase in the efficacy of anti-tick vaccines [35,36,37]. Considering this statement and the adjuvant properties described for the Bm86 nanoparticles produced by *P. pastoris* yeasts, which are the active pharmaceutical ingredient of Gavac^TM^ [38], this protein should be the candidate of choice as a carrier protein for the efficient presentation of pP0 to the host immune system. The development of pP0 as a commercial anti-tick vaccine could be a feasible substitute for KLH because of the dual role that the Bm86 protein would have in the vaccine preparation as an immunogen against ticks and also as an adjuvant. The results showed here demonstrated that Bm86 is able to enhance the immune response against pP0 and tick damages and expressed in an efficacy of around 85% against *R. sanguineus* s.l. and *R. microplus* ticks. However, under these conditions, there is not an increase in the efficacy obtained against ticks with respect to those obtained using the pP0–KLH conjugate, as could be expected of a vaccine candidate in which two antigens against ticks are combined. One explanation to this fact, suggested by our results, is that the partial conjugation of pP0 to 49 out of 54 Lys residues and the N terminal end of the Bm86 protein could eliminate, at least to some extent, important epitopes which are responsible for the protective effect of this antigen against ticks. According to our results, the magnitude of the anti-Bm86 response obtained for both groups of pP0–Bm86-immunized animals was similar to the historical response obtained in animals vaccinated with Gavac^TM^, therefore the quality of this response is what could be affected when Bm86 is used as a carrier protein of pP0 by using this chemical conjugation. Further immunization and challenge experiments should be performed, in which the intact Bm86 protein and pP0–KLH conjugate would be administered to the same animal in separate injections or mixed in the same formulation, without chemical conjugation, in order to compare their efficacies with those obtained when animals are immunized with each antigen alone and with the chemical conjugate. These experiments should answer the question if a cooperation is possible between antibodies against these two proteins. A priori, an increase in anti-tick efficacy may not be obvious by combining these antigens, because the specific functions of these proteins blocked by the antibodies are still unknown. Despite the fact that the Bm86 protein has been used as an antigen against ticks since the 1990s, its biologic function in the midgut cell membrane is so far unknown. In the case of the P0 protein, it has two very well-known biological functions: as an essential component of cell synthesis machinery and it is involved in DNA repair and apoptosis when dephosphorylated [39,40]. If either of these two functions is affected by anti-pP0 antibodies, the question about how these antibodies interact with the protein in its intracellular location remains to be clarified. Nevertheless, P0 has also been described as located on cell membranes and in tick saliva, but there its functions are still unknown [41,42,43,44,45,46]. Finally, economic considerations should be also taken into account to evaluate the results shown by this study. Authors that have obtained an efficacy increase against ticks by the combination of two or more antigens started from a baseline efficacy under 60% with an antigen alone [47,48,49]. The necessity to increase efficacy against ticks could justify the addition of a new antigen in the vaccine formulation and the cost increase to produce the vaccine [50]. However, when the efficacy of an antigen alone is more than 80%, like in this case, increasing it could be not so important. The most relevant role of the Bm86 protein obtained with this conjugation strategy was as an economically feasible adjuvant to develop pP0 as an anti-tick vaccine. This conclusion is even more validated if we consider that the addition of a new antigen in the vaccine formulation will not be enough to justify the cost increase to produce the vaccine compared with the slight effect obtained on efficacy [50] and that resistance against ticks in nature is not a knockdown effect, as for those shown by chemicals. Species genetically resistant to ticks, or with acquired resistance after repeated tick infestations, show a reduction in the number of ectoparasite infestations, keeping a tick threshold [51]. Anti-tick vaccine developers have looked for antigens to immunize susceptible hosts, converting them in tick-resistant animals. In connection with this idea, the success measure of anti-tick vaccines should be re-defined and instead of the effects of a classical vaccine, effects similar to natural resistance against ticks should be expected, for which the efficacy is not 100%. There is evidence that a vaccine against ticks with an efficacy greater than 50% is enough to reduce the tick population in a locality without eradicating them after two or three generations of these ectoparasites have fed on vaccinated hosts [52]. In our experience, the control and reduction of tick populations is possible by employing control programs with integrated management strategies whose backbone is vaccination, even if vaccine efficacy is not 100% [6,8,9]. For all these reasons, and despite the fact that in this work the use of Bm86 as a carrier protein for pP0 did not increase the efficacy of pP0 to more than 85%, it was able to enhance the specific immune response of hosts against the peptide, keeping this high efficacy against ticks that may allow the economically and technically feasible development of this peptide as a commercial vaccine.

## 4. Materials and Methods

### 4.1. Chemical Synthesis and Conjugation of pP0 Peptide to Bm86 and KLH as Carrier Proteins

The P0 peptide (NH_2_-CAAGGGAAAAKPEESKKEEAK-CONH_2_), named here as pP0, was obtained by chemical synthesis using the Fmoc strategy [53]. The pP0 corresponds to a 20 amino acid peptide located near the C-terminal region of the acidic ribosomal protein P0 of *Rhipicephalus* spp., showing the lowest sequence identity to the orthologous protein in mammals. At the same time, the reported sequence of this peptide is very conserved among tick species, showing more than 85% sequence identity among species of different tick genera and is 100% identical among species of the same genus [10]. A Cys residue was intentionally introduced at the N-terminal end in the pP0 synthesis to guarantee its chemical conjugation to the carrier protein. The purification of the synthesized peptide was performed by reverse phase chromatography in a C18 column (Zorbax 883995-902) by using an HPLC system. The eluents A (TFA/H_2_O, 0.01%) and B (TFA/acetonitrile 0.05%) were mixed in a gradient from 5 to 60% of B over 35 min. The pP0 purity was assessed by Reversed Phase—High-Performance Liquid Chromatography (RP-HPLC) and the identity was confirmed by electrospray mass spectrometry (ESI-MS).

The Bm86 protein used in the conjugation process was produced by the fermentation of a recombinant *P. pastoris* yeast clone at the CIGB of Camagüey (Gavac^TM^ active ingredient, Lot: 14.1302-4). The Bm86 protein or KLH protein (Sigma, USA) and the pP0 were linked by using the BMPS bi-functional cross-linker reagent. Briefly, the carrier protein reacted with BMPS dissolved in a dimethylformamide solution for a half an hour at room temperature. The excess of BMPS was eliminated by size exclusion chromatography using a PD-10 column from Pharmacia (GE Healthcare). The modified carrier protein containing lysine residues activated with a maleimide group reacted for 3 h at room temperature with Cys^1^pP0 dissolved in saline phosphate buffer (PBS; NaCl 7.5 g/L, Na_2_HPO_4_ 2.38 g/L and KH_2_PO_4_ 2.72 g/L, pH = 6) in a ratio (1:1) *m/m*. The excess of uncoupled Cys^1^pP0 was eliminated by an extensive membrane dialysis against PBS for 48 h with at least three buffer exchanges by using a 12–14 kDa cutoff membrane (Sigma, USA). The final concentration of conjugates was estimated using the bicinchoninic acid method [54].

### 4.2. pP0-Bm86 Deglycosylation with PNGase-F

The enzymatic deglycosylation of pP0–Bm86 conjugate using PNGase F was performed according to the manufacturer’s instructions. Briefly, the protein was denatured at 70 °C for 10 min in 0.1% SDS and 5% β-mercaptoethanol and was cooled to room temperature. Nonidet P-40 detergent (NP-40) was added to 1% of final concentration before PNGase F addition. Deglycosylation was carried out in 50 mM phosphate buffer pH 7.5, using 5 U of PNGase F/mg of glycoprotein at 37 °C for 3 h. After this, the protein was precipitated by adding three volumes of cold ethanol and standing at −20 °C for 1h followed by centrifugation at 5500× *g* for 10 min. The *N*-glycans released by the PNGase F treatment were purified by a GlycoClean H cartridge, as described by the supplier (Glyko, Novato, CA). After drying, they were labeled with 2-aminobenzamide (2-AB) according to a previously described methodology [55]. The 2-AB labeled N-glycans were then separated from the excess of reagent by ascending paper chromatography (Whatmann 3MM, Sigma-Aldrich, St Louis MO, USA) using pure acetonitrile as a mobile phase. The desalted 2-AB *N*-glycans were filtered through a 0.45 μm polytetrafluoroethylene syringe filter (Millipore, Billerica, MA, USA) and dried in a speed vacuum centrifuge (Thermo Scientific SAVANT, Pittsburgh, PA, USA). The 2-AB *N*-glycans were analyzed by matrix-assisted laser desorption/ionization mass spectrometry (MALDI-MS) in the same conditions as described previously [56].

### 4.3. SDS-PAGE Analysis and pP0 Payload Estimation of the pP0–Bm86 Conjugate

BSA, the pP0–Bm86 conjugate, the Bm86 carrier protein and the deglycosylated products of the two last proteins were analyzed by using 8% SDS-PAGE under reducing conditions. The gel was stained with Coomassie blue R-250 (Sigma, USA). A wide range molecular weight kit (Bio-Rad; USA) was used to estimate the protein molecular masses. The payload was determined by the Gel Analyzer free software available at http://www.gelanalyzer.com/ by calculating the mass difference between the deglycosylated Bm86 and the centroid of the band corresponding to the deglycosylated pP0–Bm86 conjugate and divided by the molecular mass of pP0 + linker (2.1 kDa).

### 4.4. Reduction and S-Alkylation of pP0-Bm86

The glycosylated and deglycosylated pP0–Bm86 conjugates were dissolved in 50 mM ammonium bicarbonate buffer containing 0.5 mol/L of guanidinium chloride, pH 8.3. Dithiothreitol (DTT) was added at 10 mM, and the solution was incubated at 37 °C for 2 h. A concentrated solution of acrylamide was subsequently added to reach a final concentration of 20 mM. This mixture was incubated at room temperature for 1 h. Samples were dialyzed against 50 mM ammonium bicarbonate buffer (pH 8.3) overnight at 4 °C.

### 4.5. Proteolytic Digestions of pP0–Bm86 Conjugate

Reduced and S-alkylated conjugates at 5 mg/mL in 50 mM ammonium bicarbonate buffer (pH 8.3) were digested separately with a sequencing grade trypsin (Promega, Madison, WI, USA) and *Staphylococcus aureus* V8 protease (Sigma, St. Louis, MO, USA) using 1:100 and 1:50 ratios of enzyme/substrate, respectively. The digestions proceeded overnight at 37 °C. Half of the tryptic digestion was additionally digested in tandem with V8 protease using a 1:50 ratio of enzyme/substrate at 37 °C for 4 h. The proteolytic digestions were quenched by adding 2 mL of 10% (v/v) of formic acid (FA). The resultant proteolytic peptides were desalted by SecPak cartridges (Waters, Milford, MA, USA), dried in a speed-vac (SAVANT, Buenos Aires, Argentina) and stored at −20 °C until their analysis by LC-MS/MS.

### 4.6. Liquid Chromatography–Tandem Mass Spectrometry (LC–MS/MS) Analysis

Proteolytic peptides were reconstituted in 0.2% of FA and were loaded into an enrichment precolumn (40 nL, 5 μm) packed with Zorbax 300SB-C18 (300 Å) (SmMol-Chip 43 II, Agilent Technologies) that was previously equilibrated with 0.1% (v/v) of FA solution. The peptides retained in the precolumn were desalted by washing with the same equilibration buffer for 5 min using a 2 μL/min flow delivered by an Agilent 1260 Infinity capillary pump. After finishing the desalting step, the precolumn was online coupled to the analytical reversed phase column (75 mm, 43 μm) filled with the same packing material previously mentioned. The peptides were separated with a constant flow rate of 400 nL/min by mixing buffer A (0.1% FA solution) and buffer B (90% of ACN/0.1% of FA) in a linear gradient from 3% to 70% of B for 15 min by using two Agilent 1260 Infinity Nanoflow LC pumps.

The eluted peptides were sprayed into a G6530AA accurate mass quadrupole time-of-flight (QTOF) LC/MS system (Agilent Technologies, Santa Clara, CA, USA) by using 1900 volts and 65 volts as the capillary and skimmer voltages, respectively. The high-resolution mass spectra (m/∆m~20,000) were acquired from m/z 300 to 2000 Da at 2 GHz in only MS mode in order to detect the molecular mass of proteolytic peptides. The mass spectrometer was calibrated using a mixture of polyethylene glycols (G1969-85000, Agilent Technologies).

In the collision-induced dissociation (CID) experiments, two ramps of collision energies were used to obtain fragments derived from 2+ (*m/z* 500, CE 15eV–*m/z* 1500, CE 35 eV) and 3+ (*m/z* 500, CE 10 eV–*m/z* 1500, CE 30 eV) precursor ions. Peptides with charge states higher than 3+ were fragmented under identical conditions to that described for triply charged ions. Singly charged ions were not analyzed by ESI-MS/MS in the tryptic digest. The data were acquired in positive ion mode using Mass-Hunter Data Acquisition Software B.05.00 from Agilent.

### 4.7. Identification of Conjugation Sites on pP0–Bm86 Conjugate

The LC-MS/MS runs were exported as *.mgf files using MassHunter Qualitative analysis software (Version 5.0 October 2011) from Agilent Technologies and loaded into four software programs used for the identification of cross-linked peptides: PEAKS [57], pLink [58,59], StavroX [60] and Protein Prospector [61,62]. The identification of the conjugation sites was based on the assignment of the MS/MS spectra to proteolytic peptides of Bm86 and pP0 linked through BMPS, known as intermolecular type II cross-linked peptides [19].

Two strategies were used to assign the conjugation sites. The first strategy was applied in order to allow the identification of cross-linked peptides by using PEAKS software, which is regularly used in proteomic experiments for protein identification. In this approach, type II cross-linked peptides were “treated” as linear peptides by using a transformation in which new variable modifications of lysine residues were created considering the proteolytic fragments of pP0 that could remain linked to Bm86 through BMPS. The ESI-MS/MS spectra of conjugated peptides assigned by PEAKS software were extracted as individual compounds from the raw data using MassHunter Qualitative software and they were manually inspected. Positive identifications of cross-linked peptides containing conjugation sites were accepted for those peptide spectrum matches (PSM) above 5% of the false discovery rate (FDR) that additionally showed backbone fragment ions derived from both Bm86 and pP0, as well as when fragment ions evidenced the presence of a conjugated lysine residue in the MS/MS spectrum [62]. Finally, the automatic assignment of cross-linked peptides containing conjugation sites was additionally confirmed by inspecting the presence of “reporter ions” attributed to Cys^1^pP0 proteolytic fragment ions [62].

In the second strategy, conjugation sites were directly identified from intermolecular type II cross-linked peptides using Protein Prospector (Santa Cruz Biotechnology, Inc, Santa Cruz, CA, USA), StavroX (http://www.stavrox.com/) and pLink (https://www.cog-genomics.org/plink2/) software that have been specifically developed to study protein–protein interactions in functional proteomics using the combination of cross-linking reactions and mass spectrometry. The mass error allowed for matching experimental and the expected masses of precursor and daughter ions in the assignment of Bm86 conjugation sites were 20 and 50 ppm, respectively, when StavroX, pLink and Protein Prospector software were used. The same variables and fixed modifications described for PEAKS software (Bioinformatics Solutions Inc., Waterloo, ON, Canada) were considered when the other software programs were used. A fasta sequence database containing only the Bm86 carrier protein sequence (609 amino acids) and the full-length sequence of pP0 (21 amino acids including the *N*-terminal Cys intentionally introduced in the pP0 synthesis) was loaded into these three software programs. In the cases of pLink and StavroX software, only a five percent FDR was accepted in the output list of cross-linked peptides. When Protein Prospector software was used, the cross-linked peptide assignment followed criteria suggested by authors, in which positive identification is considered only when proteolytic peptides show positive score differences and a total score higher than 15. The MS/MS spectra of all cross-linked peptides were also manually inspected. Similar retention times for MS/MS spectra assigned to the same cross-linked peptide, but with different charge states in the ESI-MS analysis, were also taken into account as additional criteria for the conjugation site identification.

PEAKS software from Bioinformatic Solutions was additionally used to determine the sequence coverage and to confirm the sequence identity of the Bm86 carrier protein by inserting its amino acid sequence into a database containing 5255 protein sequences of *P. pastoris* yeast deposited on the Uniprot website (https//www.uniprot.org/). The new database was validated and exported as a decoy database to estimate the false discovery rate (FDR). The FDR was set to 5% for the peptide spectrum matching during the protein identification process. The mass error for peptides and daughter ion assignments was set at 0.05 Da. The spider option, available in PEAKS software, was used to explore in depth the presence of more than 500 post-translational modifications (PTMs) and amino acid changes in the carrier protein sequence.

### 4.8. Bioinformatic Analysis

Cross-linked peptides identified by the four evaluated software programs were compared in order to get information about the redundancy degree and the identification reliability. Moreover, overlapping information about conjugation sites provided by the four software programs was evaluated by Venn diagrams created using Venny Software v 2.1 (http://bioinfo.gp.cnb.csic.es/tools/venny/). The sequence coverage map of the pP0–Bm86 conjugate was obtained by considering all proteolytic peptides obtained by the LC-MS/MS analysis, being cross-linked, or not, to pP0.

### 4.9. Immunization and Challenge Experiment in Dogs

All procedures involving animals and samplings were carried out in accordance with the Guide for the Care and Use of Laboratory Animals [63] and were approved by the Ethic Committees at the institutions where the experiments were carried out (010221/13 in Brazil and 01/16 in Cuba).

Twelve beagle dogs at an average of 4 years old and weighing around 10 kg were placed randomly in three experimental groups of four animals each at the Department of Veterinary Pathology, Faculty of Agronomic and Veterinary Sciences, São Paulo State University, Jaboticabal campus, Brazil. Dogs were fed with the Veg Dog commercial pellet diet (VeggiePets, 15 Kg, UK) and water ad libitum. The immunization by subcutaneous injection was performed with 1 mL of oily preparations using Montanide ISA 50 (SEPPIC, France) in a proportion 60/40 of immunogen/adjuvant on days 0, 21 and 36. Group 1 received the pP0–KLH conjugate in a dose of 500 μg/animal and group 2 the pP0–Bm86 conjugate in the same dose. Group 3 was the negative control injected with PBS in the same oily formulation. The general behavior and body temperature of the animals were monitored daily throughout the test.

Serum samples of all dogs were taken on days 0, 21, 36, 51, 68 and 87 in order to measure the antibody responses by indirect ELISA. KLH, Bm86 and pP0 were used to coat ELISA plates overnight at 4 °C. Sera were serially diluted 1:2 with PBS. The plates were incubated with the diluted sera for 1 h at 37 °C and then incubated with 1:10,000 anti-dog IgG–HRP conjugate (Sigma, USA) for 1 h at 37 °C. The color reaction was developed with a substrate solution containing 0.4 mg/mL of o-phenylenediamine in 0.1 M citric acid and 0.2 M Na_2_HPO_4_, pH 5.0, and 0.015% of hydrogen peroxide. The reaction was stopped with 2.5 M H_2_SO_4_ and the OD 490 nm was determined. The antibody titer was established as the reciprocal of the highest dilution, at which the mean OD of the study serum was three times the mean OD of the negative control serum. Results were presented as the geometric mean of each group. The data were base 10 log transformed to compare antibody titers against pP0 between the two immunized groups by using *t*-tests performed in Prism (version 6.0 for Windows; GraphPad Software, USA).

Each dog was challenged 15 days after receiving the last booster vaccination with 300 ± 25 larvae, 150 ± 10 nymphs and 50 adults (30 females and 20 males) of *R. sanguineus* ticks from a tropical lineage maintained at the FCAV-UNESP, Jaboticabal, Brazil [64]. Each tick stage was fed inside of an individual craft feeding chamber glued to the shaved flank of dogs. Elizabethan collars were used to prevent the chamber removal. All collected feed ticks were counted and kept in an incubator at 27 ± 1 °C with 80% relative humidity, and a photoperiod of 12 h of light. The detached engorged females were placed into individual plastic vials and larvae and nymphs were stored in groups of daily batches.

The mortality in the molting process of larvae and nymphs was recorded and also the female mortality in the oviposition period. Egg masses were weighed and larval hatching was also recorded. Group averages for each measured parameter were compared by ANOVA and Bonferroni multiple comparison tests performed in Prism (version 6.0 for Windows; GraphPad Software, USA). The overall efficacy of each antigen (E) was calculated, including the effects on each tick stage, as E = 100 × (1 − [RL × VL × RN × VN × RA × PA × FE]) where RL and VL represent the effects of each immunogen on the larva yield and mortality in the molting process compared to the control group, respectively. RN and VN are the effects of each immunogen on the nymph yield and mortality in the molting process compared to the control group, respectively. RA and PA are the effects of each immunogen on female recovery and oviposition compared to control group, respectively. FE is the effect of each immunogen on egg fertility. It was calculated as the ratio between the hatching percentages of eggs laid by ticks fed on vaccinated animals compared to the control group.

### 4.10. Histology and Ultrastructure of Ovaries from Ticks Fed on Immunized Dogs

The first fifteen full-fed females detached from dogs in each experimental group were dissected under a stereomicroscope to collect ovaries. Five randomly selected ovaries were used for histological studies. These ovaries after washed in PBS were placed in 4% paraformaldehyde fixative solution for 24 h. They were then transferred to PBS again for 24 h and dehydrated in a series of baths with increasing concentrations of ethyl alcohol (70%, 80%, 90% and 95%) for 15 min each. Soon after, materials were soaked and included in HistoResin (Leica, Wetzlar, Germany) by using plastic molds. The blocks were glued on wooden supports for sectioning in a rotatory microtome, from which cuts with a thickness of 3 μm were obtained and collected on glass slides. After drying, slides were stained with hematoxylin–eosin (HE) according to histological routine and assembled with Canada balsam on a coverslip for later observation and photographic documentation under a light microscope (MOTIC BA 300) coupled to a computer. Three random sections from each female in each group were selected for yolk granule quantification. The number of yolk granules inside oocytes in the histological sections was counted and the group averages were compared by ANOVA and Bonferroni multiple comparison tests performed in Prism (version 6.0 for Windows; GraphPad Software, San Diego, CA, USA).

Another five ovaries from each experimental group were placed in a 2.5% glutaraldehyde fixing solution. After that, they were dehydrated in a bath series with increasing concentrations of acetone (70%, 80%, 90%, 95% and 100%) for 15 min each. Finally, the ovaries were fixed on a metal support with double adhesive tape and bombarded with gold and carbon. They were analyzed and photographed with a JEOL Scanning Electron Microscope (model JSM 5410). The numbers of mature oocytes and immature oocytes adhered to the ovary wall in each sample were recorded and compared by ANOVA and Bonferroni multiple comparison tests performed in Prism (version 6.0 for Windows; GraphPad Software).

Ultra-morphology studies were performed using the remaining five dissected ovaries of engorged females from each experimental group. Ovaries were placed in a 2.5% glutaraldehyde fixing solution. At that point, they went through two 15-min washes each, in 0.1 M sodium cacodylate buffer solution, and were subsequently post-fixed in a 1% osmium tetroxide solution for 2 h. Again, they were washed twice with 0.1M sodium cacodylate for 15 min each. Next, the ovaries underwent a 10% alcohol bath for 15 min and were subsequently contrasted in 2% uranyl acetate plus 10% ethanol for 2 h. Subsequently, dehydration was carried out in a 10-min bath series of acetone of increasing concentrations (50%, 70%, 90%, 95% and 100%). Soon after, materials were immersed in an acetone and resin mixture in a 1:1 proportion, where they remained for 12 h. Finally, the ovaries were covered in pure Epon Araldite resin with a catalyst in an oven at 60 °C for 24 h. All materials were sectioned in an ultramicrotome and ultrafine sections were collected on copper grids. They were contrasted with uranyl acetate and lead citrate for 45 and 10 min, respectively. The grids containing the sections were examined and photographed under an electron transmission microscope (Philips CM 100, Philips, Amsterdam, The Netherlands).

### 4.11. Immunization and Challenge Experiment in Cattle

Fifteen naive bovines from the Cuban Institute of Animal Science (ICA), aged between 10 and 12 months, of the Cuban Siboney breed (5/8 Holstein and 3/8 Cebu) and weighing around 100 Kg were randomly distributed in three experimental groups of five animals in each one. Bovines were immunized by deep intramuscular injection in the neck region with 2 mL of pP0–KLH, pP0–Bm86 and PBS oily preparations, respectively, using Montanide ISA 50 in the same proportions and following the same schedule and doses as in the dog experiment. The trial was conducted in the ICA facilities with animals individually housed with relative mobility, receiving daily forage and water ad libitum and 2.5 kg of a concentrated food, which contained 86% dry matter, 11.84% crude protein and 10.8 MJ of energy by kg of dry matter. During the study, weights and blood parameters were determined for each animal as health indicators. Serum samples of bovines were taken on days 0, 21, 36, 51, 86 and 170 in order to measure the antibody responses by the indirect ELISAs described previously. In these ELISAs, a non-related pP0 chemical conjugate was used in the plate coating to measure the specific response against pP0, and in all cases, an anti-bovine IgG–HRP conjugate (Sigma) was used as a secondary antibody. Fifteen days after the last immunization, each bovine was progressively challenged with 3000 larvae (1000 larvae/day) in three feeding chambers of 1000 larvae each of *R. microplus* ticks from a colony established in the Cuban National Laboratory of Parasitology from the Cayo Coco field isolate [10]. Detached engorged females were collected, individually weighed and immobilized in a disposable Petri dish during oviposition in the incubator, using the conditions described previously. The egg mass weights were also individually recorded. Egg mass pools of detached females from the same animal on the same day were kept until hatching, which was determined by percentage by visual observations [65]. The statistical analyses of all recorded data were performed by ANOVA and Bonferroni multiple comparison tests in Prism (version 6.0 for Windows; GraphPad Software). The overall efficacy of each antigen (E) was defined by percentage as E (%) = 100 × [1 − (RA × PA × FE)] where RA, PA and FE are parameters related to engorged females with the same previous definitions as when used in the three-host tick formula. Parameters in vaccinated groups that were not statistically different compared to the control group were not considered in any efficacy calculations.

## Figures and Tables

**Figure 1 pathogens-09-00513-f001:**
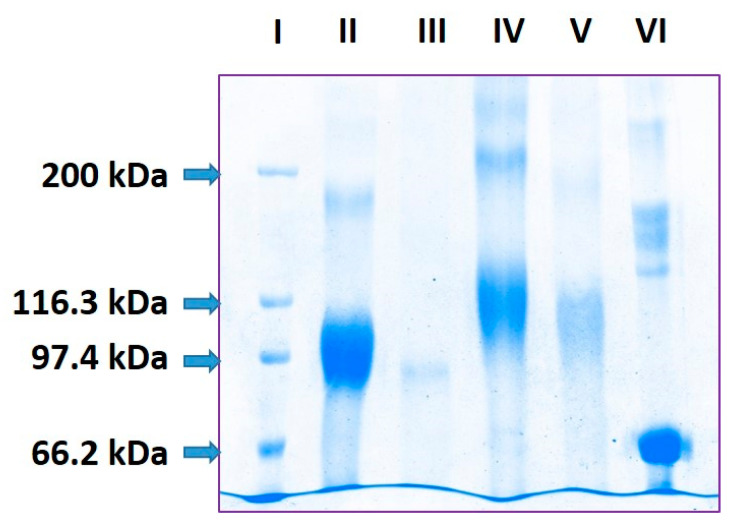
SDS-PAGE (8%) analysis under reducing conditions of the Bm86 carrier protein and the pP0–Bm86 conjugate. Lanes: (I) wide-range molecular weight marker; (II) *N*-glycosylated Bm86 protein expressed by recombinant *P. pastoris* yeast clone; (III) deglycosylated Bm86 protein with PNGase F; (IV) pP0–Bm86 conjugate; (V) deglycosylated pP0–Bm86 conjugate with PNGase F; (VI) 67 kDa molecular weight standard protein (Bovine Serum Albumin).

**Figure 2 pathogens-09-00513-f002:**
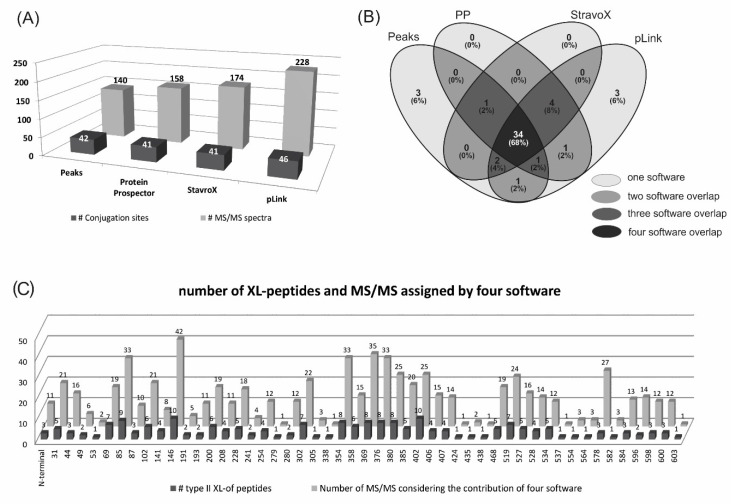
(**A**) Bars in gray represent the number of MS/MS spectra assigned by each software to intermolecular type II cross-linked peptides in the LC-MS/MS analysis of pP0–Bm86 conjugate digested with several specific proteases. Dark gray bars indicate the number of conjugation sites identified by each software; (**B**) Venn diagram showing the overlapped information provided by software used for the identification of the conjugation sites in pP0–Bm86. PP means Protein Prospector software. (**C**) Numbers on x axis represent the position of the forty-nine lysine (K) residues and the N-terminal end of the Bm86 amino acid sequence that were identified as conjugation sites for pP0. K residues located at positions 180, 431, 484, 495 and 505 were not included because no conjugation evidence was found at all. Dark gray bars indicate the number of intermolecular type II cross-linked peptides (XL) identified by all software for each conjugation site. The clear gray bars indicate the number of MS/MS spectra that supported the assignment of each conjugation site considering the total contribution of all software.

**Figure 3 pathogens-09-00513-f003:**
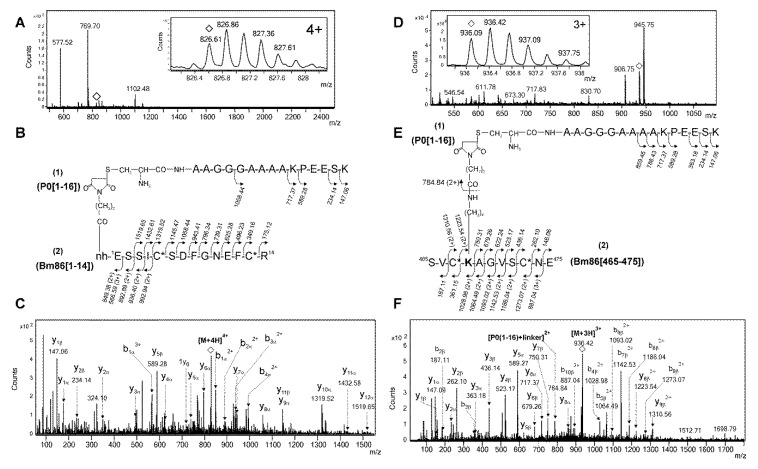
(**A**) Electrospray mass spectrometry (ESI-MS) spectrum of a fraction derived from the tryptic digestion of pP0–Bm86 conjugate that shows the isotopic distribution of 4+ ion detected at *m/z* 826.61 and labeled with an empty rhombus. (**B**) Two cross-linked peptides labeled as (1) and (2) corresponding to tryptic peptides [1–14] and [1–16] derived from N-terminal end of Bm86 carrier protein and Cys^1^-pP0, respectively. (**C**) ESI-MS/MS spectrum of the intermolecular type II cross-linked peptides labeled with an empty rhombus in (**A**). The heterobifunctional cross-linker *N*-β-(maleimidopropyloxy) succinimide ester (BMPS) appeared, linking the N-terminal end of Bm86 (^1^E) and Cys^1^ of pP0. (**D**) ESI-MS spectrum of a fraction derived from the tandem digestion with trypsin and V8 of the reduced and S-alkylated pP0–Bm86 conjugate that shows the isotopic distribution of the triply charged (3+) ion detected at *m/z* 936.09 and labeled with an empty rhombus. (**E**) Sequences labeled as (1) and (2) correspond to the proteolytic peptides [465–475] and (1–16), derived from Bm86 carrier protein and pP0 peptide, respectively. (**F**) ESI-MS/MS spectrum of two intermolecular type II cross-linked peptides shown in (**E**). The heterobifunctional cross-linker BMPS appeared, linking the K^468^ in Bm86 and Cys^1^ in pP0. C* and bold K represent a cysteine modified with propionamide and the conjugation site, respectively. The assignment of fragment ions in the MS/MS spectra shown in (**C**,**F**) is in agreement with the sequences of type II cross-linked peptides of (**B**,**E**) and with the nomenclature proposed for the fragmentation of this kind of peptides [19].

**Figure 4 pathogens-09-00513-f004:**
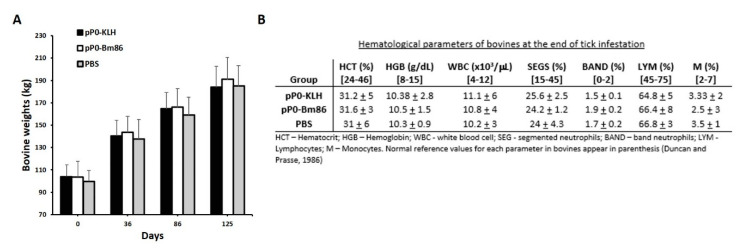
(**A**) Bovine weight average in each group during the immunization experiment and the challenge with *R. microplus* ticks. Standard deviations are represented by error bars in the positive direction. (**B**) Average ± standard deviation of hematological parameters of bovines in each experimental group at day 86 from the first immunization.

**Figure 5 pathogens-09-00513-f005:**
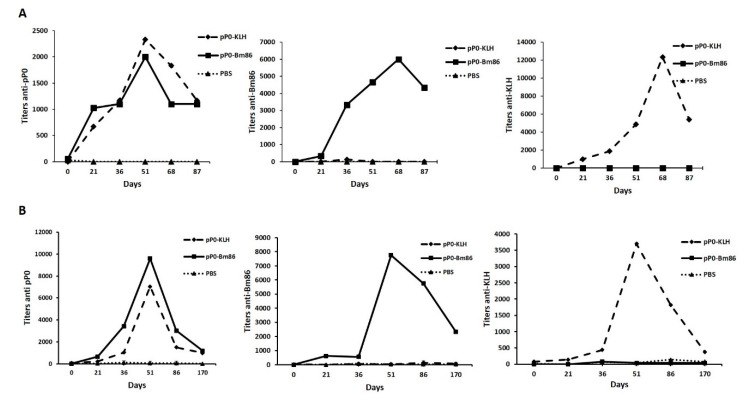
Specific antibody responses against vaccine antigens obtained in dogs (**A**) and in bovines (**B**). These antibody responses were measured by indirect ELISA and are expressed as the reciprocal of the last serum dilution with an OD at 490 nm three times greater than the OD at 490 nm average of negative sera. Each point represents the geometric mean of antibody titers in the group.

**Figure 6 pathogens-09-00513-f006:**
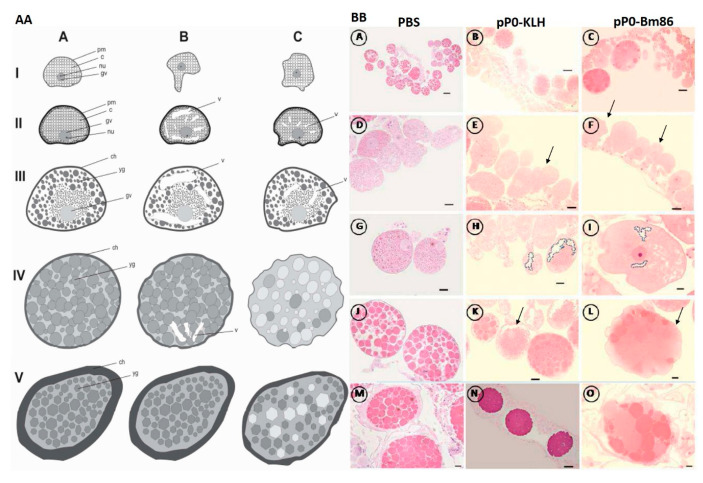
Histological changes in oocytes of *R. sanguineus* females fed on immunized dogs. (**AA**) Schematic representation, where (**A**) represents normal oocyte development as in the control group (PBS); (**B**,**C**) represent histological changes that appear in oocytes from females fed on pP0–KLH and pP0–Bm86 immunized dogs, respectively; I = oocyte I; II = oocyte II; III = oocyte III; IV = oocyte IV; V = oocyte V; pm = plasmatic membrane; nu = nucleolus; gv = germinal vesicle; c = cytoplasm; yg = yolk granules; ch = chorion; v = vacuolization. (**BB**) Histological sections of *R. sanguineus* females; (**A**–**C**) show pre-vitellogenic and vitellogenic oocytes from three experimental groups; (**D**–**F**) show the morphology of pre-vitellogenic oocytes with abnormal shapes pointed out in E and F; (**G**–**I**) show oocytes III with cytoplasmic vacuolization of oocytes from immunized groups (dotted) in G and H; (**J**–**L**) show vitellogenic oocytes where vacuolization and abnormal shapes are also evident in K and L; (**M**–**O**) show details of mature oocytes from females of three experimental groups; Bars of scale: A, D, G, J, M = 200 μm; B, C, N = 100 μm; E, F, H, K = 50 μm; I, L, O = 20 μm.

**Figure 7 pathogens-09-00513-f007:**
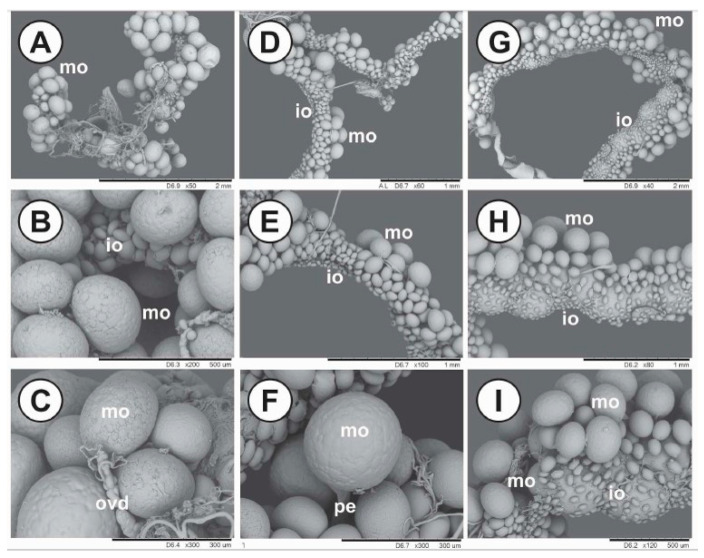
SEM of ovaries from *R. sanguineus* females fed on dogs belonging to the control group (**A**–**C**), the P0–KLH conjugate-immunized group (**D**–**F**) and the P0–Bm86 conjugate-immunized group (**G**–**I**). mo = mature oocytes; io = immature oocytes, ovd = oviduct; pe = pedicel. Scale bar: A, G = 2 mm; D, E, H = 1 mm; B, I = 500 μm; C, F = 300 μm.

**Figure 8 pathogens-09-00513-f008:**
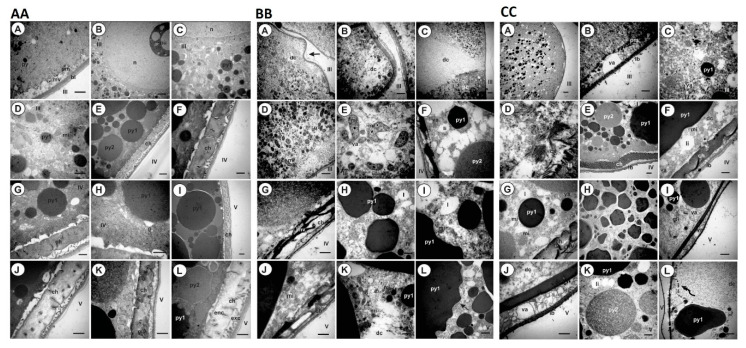
Ovary ultrastructure of the *R. sanguineus* females fed on dogs of the control group (**AA**); the pP0–KLH-immunized group (**BB**) and the pP0–Bm86-immunized group (**CC**). III = oocyte III; IV = oocyte IV; V = oocyte V; py, py1, py2 = protein granules with different electron densities; li = lipid granule; pm = plasmatic membrane; bl = basal lamina; mv = microvilli; n = nucleus; nu = nucleolus; mi = mitochondria; mf = myelin figure; g = Golgi complex; ch = chorion; exc = exochorion; enc = endochorion; va = autophagy vacuole; dc = degradation of cytoplasm; arrows = irregularly shaped granule; irregular arrows = degradation track of yolk granule. Scale bars: (**AA**)—A = 3 μm; B = 5 μm; C, D, F, G, H, J, K, L = 1 μm; E, I = 2 μm. (**BB**)—A = 2 μm; B, E–K = 1 μm; C = 5 μm; D = 3 μm. (**CC**)—A, H, L = 5 μm; B, C, D, F, G, J, K = 1 μm; E = 2 μm, I = 3 μm.

**Figure 9 pathogens-09-00513-f009:**
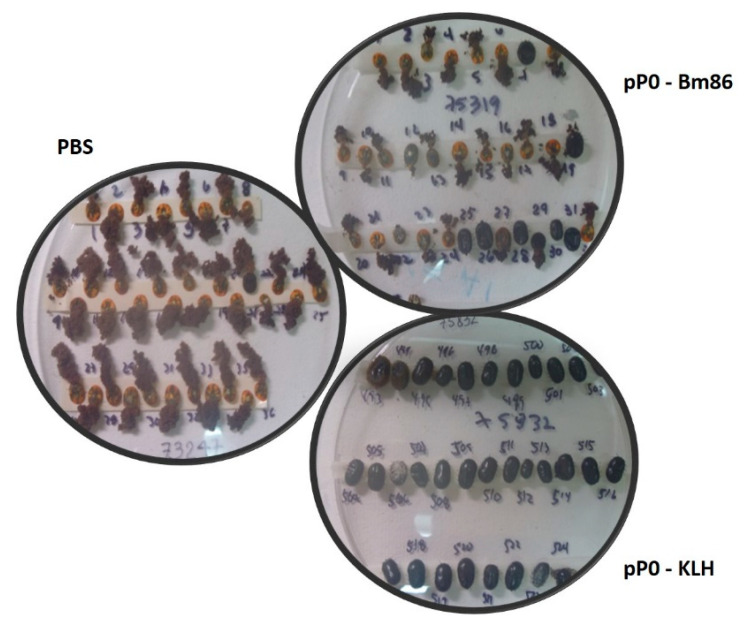
Engorged females of *R. microplus* detached from bovines of different experimental groups. Normal oviposition was observed in females from control bovines that received PBS injection. Rare and abnormal oviposition was observed for many engorged females fed on pP0–Bm86-immunized bovines and no oviposition at all was observed for many females from pP0–KLH-immunized bovines.

**Figure 10 pathogens-09-00513-f010:**
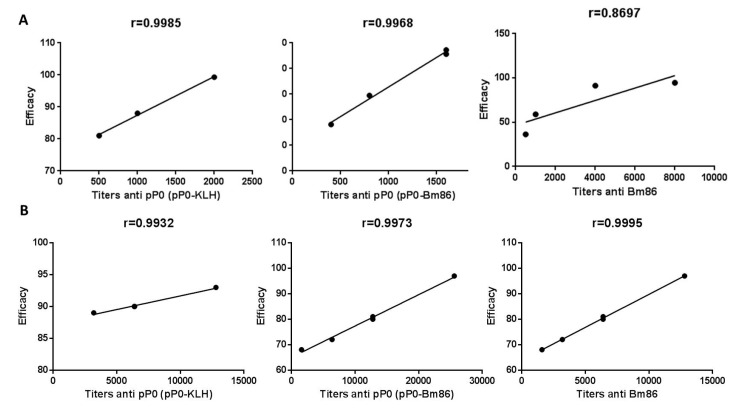
Correlation analysis between specific antibody responses and efficacy of vaccine antigens performed with Prism (version 6.0 for Windows; GraphPad Software, USA). Pearson r coefficients appear on top of each graphic. Panel **A** corresponds to dog experiment. Correlations were significant only for anti-P0 antibodies (*p* < 0.05 and *p* < 0.01 for pP0–KLH and pP0–Bm86, respectively. Panel **B** corresponds to cattle experiment. In all cases, the correlations were strongly significant (*p* < 0.001).

**Table 1 pathogens-09-00513-t001:** Effect of dog vaccination with pP0 conjugates on *R. sanguineus* infestation.

**Reduction Percent with Respect to Control of**				
**Groups**	**Larval Yield**	**Larval Viability (%)**	**Nymph Yield**	**Nymphs’ Viability (%)**
**pP0-KLH (*n* = 4)**	15% (167 ± 75) ^a^	0% (95 ± 2) ^a^	63% (40 ± 15) ^b^	47% (50 ± 4) ^b^
**pP0-Bm86 (*n* = 4)**	30% (138 ± 61) ^a^	0% (92 ± 5) ^a^	38% (67 ± 21) ^b^	33% (64 ± 9) ^b^
**PBS (*n* = 4, Control)**	(196 ± 72) ^a^	(92 ± 3) ^a^	(108 ± 15) ^a^	(95 ± 6) ^a^
**Reduction Percent with Respect to Control of**				
**Groups**	**Female Yield**	**Egg Mass Weight (mg)**	**Hatchery (%)**	**E (%)**
**pP0-KLH (*n* = 4)**	54% (11 ± 8) ^b^	22% (51.01 ± 8.4) ^b^	28% (68 ± 7) ^b^	**95%**
**pP0-Bm86 (*n* = 4)**	38% (15 ± 3) ^b^	5% (62.06 ± 25.0) ^a,b^	9% (86 ± 3) ^b^	**86%**
**PBS (*n* = 4, Control)**	(24 ± 6) ^a^	(64.93 ± 18.06) ^a^	(94 ± 5) ^a^	

E (efficacy) is calculated as 100 × [1 − (RL × VL × RN × VN × RA × PA × FE)] where RL, VL, RN, VN, RA, PA and FE represent the immunogen effect on larva yield, larva viability, nymph yield, nymph viability, female yield, egg mass weight and egg fertility, respectively. Only parameters showing statistically significant differences compared to those in the control group were included in E calculation. In parentheses, the average ± SD of each recorded parameter are shown. Different letters ^a^ and ^b^ mean statistically different groups (ANOVA, Bonferroni multiple test, *p* < 0.05).

**Table 2 pathogens-09-00513-t002:** Effect of bovine vaccination with pP0 conjugates on *R. microplus* infestation.

Reduction Percent with Respect to Control of				
Groups	Female Yield	Egg Mass Weight (mg)	Hatchery (%)	E (%)
**pP0–KLH (*n* = 5)**	84% (102 ± 24) ^b^	7% (72.89 ± 51.94) ^a,b^	28% (71 ± 10) ^b^	**89%**
**pP0–Bm86 (*n* = 5)**	72% (177 ± 56) ^b^	22% (69.36 ± 44.81) ^b^	34% (65 ± 18) ^b^	**84%**
**PBS (*n* = 5, Control)**	(639 ± 222) ^a^	(78.29 ± 44.84) ^a^	(98 ± 5) ^a^	

E (efficacy) is calculated as 100 × [1 − (RA × PA × FE)] where RA, PA and FE represent the immunogen effect on female yield, egg mass weight, and egg fertility, respectively. Only parameters showing statistically significant differences compared to those in the control group were included in E calculation for each vaccine candidate. In parentheses, average ± SD of each recorded parameter are shown. Different letters ^a^ and ^b^ mean statistically different groups (ANOVA, Bonferroni multiple test, *p* < 0.05).

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
