# Peer review of "Functional and Mass Spectrometric Evaluation of an Anti-Tick Antigen Based on the P0 Peptide Conjugated to Bm86 Protein"

_pathogens, 2020, doi:10.3390/pathogens9060513_

Round 1
Reviewer 1 Report
Dear Author/s,
The submitted manuscript Pathogens-840028 is a detailed study related to molecular and functional characterization of anti-tick antigen pP0 peptide conjugated to Bm86 and check its efficacy as a novel vaccine target.
Functionally, its efficacy was checked on dogs and some cattle but not on human Lyme disease related pathogens. Hence in my opinion title should be modified or should include the names. Or else in broad and simple manner, it could be presented as functional and proteomic characterization of….
In introduction from line number 53 onward (first paragraph) the flow of information has broken, continue the story with some missing links which is deprived here. The same kind of problem is occurring in discussion section. Take care of it.
Methodology is very much in detail, try to make it concrete. Check the spelling, English grammar and formatting one more time at your end.
Although, it is clear that author has generated and used original dataset but some of the constituents which have already been characterized and analysed/published before with some other aspects out of it (ref no. 10, 11, 12). Avoid the repetition of results, if any.
Even though, I appreciate the efforts made by authors/acknowledging staffs to work on animal model and at laboratory level and derive the finding in contextual manner.
All the Best.
Reviewer 2 Report
The studies presented in this manuscript concern the development of an improved vaccine against tick infestation in cattle and dogs. These studies fall within the scope of the journal. The authors evaluated the efficacy of a peptide of the P0-ribosomal protein when coupled to either Keyhole Limpet Hemocyanine (KLH) or recombinant Bm86 midgut protein from Rhipicephalus microplus ticks. The authors present a detailed analysis of the biochemical structure of the conjugated P0-Bm86 protein, immunogenicity and some safety data from immunized cattle and dogs. Moreover, light- and electron microscopical data is presented, which gives a very complete picture of the effects of vaccination on female ticks. These data can be considered new and original.
The P0 peptide corresponds to a 20 amino acid peptide located nearby of the C-terminal region of the acidic ribosomal protein P0 of Rhipicephalus spp. Because the P0 protein is also present in other species (although sequence similarity with the orthologous protein in mammals is low) it is of importance to give information about the heterogeneity in R. appendiculatus and R. microplus that are used in these studies.
The recombinant protein and relevant control protein are formulated with an oil adjuvant (water in oil emulsion). It is stated that a single dose of vaccine contains 500 mg of protein (Lines 671-672). That is excessively high. Even if it were 500 µg of protein, then it would still be relatively high, given the fact that the amount of Bm86 protein in the commercial product is normally 100 µg per dose as described in Vargas et al. 2010 (ref. 32): Animals were immunized by using a deep intramuscular injection, 21 × 11/2" needles and 2 mL of Gavac PLUS, containing 100 μg of recombinant Bm86 antigen emulsified in the oil-based adjuvant Montanide 888.The authors are requested to comment on this.
In figure 5 the specific antibody responses upon immunization of dogs and bovines are presented. Titres are presented as end-point dilutions (the reciprocal of the dilution at which the reaction becomes negative). Next, the average of these titres is calculated arithmetically for each experimental group. This is not correct because these data are not normally distributed. Data should be presented as a log value (usually 2log values, or titre steps), which yields a normal distribution of the antibody titres that can easily be statistically analysed using for instance ANOVA. This should be corrected.
The authors calculated an overall value of efficacy using different parameters. This is not scientifically correct and should be calculated and presented in a more thorough way.
-In the dog study, the effect of vaccination was determined against infestation with either larvae, nymphs or adult ticks separately. From the reduction in numbers of ticks from different stages and the viability of those ticks an overall effect was calculated. This assumes that the different parameters that are used in the equation are independent of each other. That does not have to the case. For instance, suppose that 30% of the tick strain that is used has phenotype SS that defines sensitivity to the effects of vaccination, then 30% of the ticks with such phenotype will be killed on the vaccinated animals. This can be determined using larvae, nymphs and adults, as is done in these studies. However, if one infests the dogs with larvae and let these develop to nymphs, and subsequently to adults on that same dog, the overall effect of vaccination is 30% because the ticks with the SS phenotype will be killed at the larval stage and the surviving phenotype will not contain SS individuals anymore. In conclusion, the authors should give the range of efficacy assuming the parameters are dependent or independent of each other. In this case the range will vary between 30-95%.
-In the bovine study in which the one-host tick Rhipicephalus microplus was used, this effect of heterogeneity in the tick population does not play a role as the larvae develop to adult ticks at the same animal. However, also here the interpretation of the effect of vaccination on tick development is highly skewed towards effects on females and the fecundity. The origin of this problem resides in the fact that efficacy is determined by counting engorged female ticks and fecundity. These parameters are the end result of an overly complex interaction of male and female ticks on the (immunized) host. To reach full engorgement, female ticks have to interact with male ticks. If this interaction is affected, which could be due to an effect of immunization on the viability and/or fertility of the male ticks, the number fully engorged adult female ticks is affected. In addition, in the case the interaction of male and female ticks on the animal is not affected but the fertility of the male ticks is, then female ticks could reach full engorgement but might not produce viable progeny. These are just a few examples of any possible effect of vaccination. Hence, the authors should discuss the efficacy of immunization more balanced considering possible effects on the male ticks as well.
The calculated efficacy of the vaccine in cattle appeared to be correlated with the antibody titres against P0 and Bm86 (Figure 10). The authors do not present the correlation between the antibody titres against KLH and protection. Judging from the antibody dynamics that are presented in Figure 5, it is highly likely that there is a significant correlation. This evaluation should be added, and the results should be discussed considering the antibody titres against KLH as well.
The authors do not present the possible correlation between specific antibody titres in sera of dogs and the observed level of protection. This should be added.
